# Genome-Wide Association Study of Milk Composition in Karachai Goats

**DOI:** 10.3390/ani14020327

**Published:** 2024-01-21

**Authors:** Marina Selionova, Vladimir Trukhachev, Magomet Aibazov, Alexander Sermyagin, Anna Belous, Marianna Gladkikh, Natalia Zinovieva

**Affiliations:** 1Subdepartment of Animal Breeding, Genetics and Biotechnology, Russian State Agrarian University—Moscow Timiryazev Agricultural Academy, Timiryazevskaya Street, 41, 127434 Moscow, Russiamarianna1001@yandex.ru (M.G.); 2L. K. Ernst Federal Research Center for Animal Husbandry, Dubrovitsy 60, 142132 Podolsk, Moscow Region, Russia; priemnaya-vij@mail.ru (M.A.); alex_sermyagin85@mail.ru (A.S.); abelous.vij@ya.ru (A.B.); n_zinovieva@mail.ru (N.Z.)

**Keywords:** genome-wide association study, milk composition, Karachai goats

## Abstract

**Simple Summary:**

The growth rate of dairy goat farming is due to the special qualities of goat milk, which are directly determined by its composition. Goat’s milk has a higher concentration of special fatty acids but less protein than cow’s milk, which is why high-quality cheeses are made from goat’s milk. Therefore, the goal of breeding dairy goats is not only to increase their milk yield but also to improve the quality composition of their milk. Karachai goats are a breed of local selection, the milk of which is widely used in the regions of the North Caucasus for the production of unique cheeses. While a relatively large number of genome studies have been carried out for transboundary goat breeds, such as Saanen goats, there is not yet enough research on local goat breeds. That is why we conducted a genome-wide association study (GWAS) on the milk component traits of Karachai goats. The use of genomic evaluation in the selection of Karachai goats will increase the rate of breeding progress in terms of enhancing milk productivity traits and thereby increase the profitability of their use.

**Abstract:**

This study is first to perform a genome-wide association study (GWAS) to investigate the milk quality traits in Karachai goats. The objective of the study was to identify candidate genes associated with milk composition traits based on the identification and subsequent analysis of all possible SNPs, both genome-wide (high-confidence) and suggestive (subthreshold significance). To estimate the milk components, 22 traits were determined, including several types of fatty acids. DNA was extracted from ear tissue or blood samples. A total of 167 Karachai goats were genotyped using an Illumina GoatSNP53K BeadChip panel (Illumina Inc., San Diego, CA, USA). Overall, we identified 167 highly significant and subthreshold SNPs associated with the milk components of Karachai goats. A total of 10 SNPs were located within protein-coding genes and 33 SNPs in close proximity to them (±0.2 Mb). The largest number of genome-wide significant SNPs was found on chromosomes 2 and 8 and some of them were associated with several traits. The greatest number of genome-wide significant SNPs was identified for crude protein and lactose (6), and the smallest number—only 1 SNP—for freezing point depression. No SNPs were identified for monounsaturated and polyunsaturated fatty acids. Functional annotation of all 43 SNPs allowed us to identify 66 significant candidate genes on chromosomes 1, 2, 3, 4, 5, 8, 10, 13, 16, 18, 21, 23, 25, 26, and 27. We considered these genes potential DNA markers of the fatty acid composition of Karachai goat milk. Also, we found 12 genes that had a polygenic effect: most of them were simultaneously associated with the dry matter content and fatty acids (METTL, SLC1A 8, PHACTR1, FMO2, ECI1, PGP, ABCA3, AMDHD2). Our results suggest that the genes identified in our study affecting the milk components in Karachai goats differed from those identified in other breeds of dairy goats.

## 1. Introduction

In the last decade, dairy goat farming has rapidly developed in Russia. For example, in 2010, out of the total population of 2.1 million dairy and dairy–meat goats, there were only about 340 thousand, whereas in 2022, out of 1.9 million, there were around 750 thousand, which is 2.2 times more [1]. This trend corresponds with global tendencies toward increasing the number of dairy goats and goat milk production [2,3]. The interest in dairy and dairy–meat goat farming worldwide is driven by the stable demand for environmentally friendly and natural products, including goat milk, which possesses a number of valuable functional properties. It is known that goat milk, unlike cow milk, is characterized by a high dispersion of fat globules, a high content of unsaturated short-chain fatty acids and β-casein, an extremely small amount of alpha-1s-casein (which causes allergic reactions to cow milk), and virtually no monosaccharides. These characteristics determine the dietary properties of goat milk and its advantage in producing high-quality cheeses [4].

Genome-wide association studies (GWASs) are widely used to identify the genomic regions that determine economically valuable traits in farm animals and incorporate them into breeding programs [5]. For goats in general, and dairy goats in particular, there have been significantly fewer studies compared to dairy cattle [6,7,8]. Nevertheless, there are certain results from research conducted on different breeds of dairy goats in various countries [9,10,11]. For instance, a GWAS has been conducted and candidate genes for milk production traits in Saanen and Alpine goats have been identified in France (Martin et al., 2016, 2017, 2018 [12,13,14]; Talouarn et al., 2020 [15]), the United Kingdom (Mucha et al., 2018 [16]), New Zealand (Scholtens et al., 2020 [17]), Canada (Massender et al., 2023 [18]), and America (Tilahun et al., 2020 [19]). It was found that a significant number of SNPs (single nucleotide polymorphisms) were located on chromosomes 14, 19, and 29. These SNPs were found both in known “milk” genes that have been identified as significant for dairy cattle selection, as well as in new candidate genes. For example, the known genes include DGAT1 (Diacylglycerol O-Acyltransferase 1), ABCG2 (ATP-binding cassette sub-family G member 2), ADAMTS20 (ADAM Metallopeptidase with Thrombospondin Type 1 Motif 20), and PAPPA2 (pregnancy-associated plasma protein A2). The candidate genes include ACACA (acetyl-CoA carboxylase alpha), BTN1A1 (butyrophilin subfamily 1 member A1), LPL (lipoprotein lipase), SCD (stearoyl-CoA desaturase), and SLC27A1 (Solute Carrier Family 27 Member 1). 

The tourism industry is actively developing in the North Caucasus, which, in turn, stimulates an increase in the volume of dairy products with improved functional characteristics, particularly from goat milk. Karachai goats belong to the dairy–meat direction of productivity and are widely spread in this region. They efficiently utilize alpine and subalpine pastures, known for their exceptionally rich flora. However, due to steep slopes and rocky outcrops, these pastures are often inaccessible to other types of livestock [20].

Although genome-wide association studies (GWASs) on goats have been conducted in many countries and on different breeds, the loci associated with quantitative and qualitative traits of milk production remain largely unknown (Saleh, A.A., 2022 [21]). This applies to the study of milk production in Karachai goats as well, which justifies the relevance of this research.

The objective of this study was to identify candidate genes associated with milk production traits in Karachai goats based on a genome-wide analysis of associations.

## 2. Materials and Methods

### 2.1. Sample Collection and Phenotypic Measurements

For the study, a random sampling method was used to select 167 lactating Karachai goats (from peasant/farmer households—KFH “Pyatigorsky” in the Stavropol region and KFH “Maysky” in the village of Kyzyl-Kala, the Karachai-Cherkess Republic), from which ear tissue or blood samples were obtained for DNA extraction. 

Milk samples were individually collected and preserved using MicroTabs tablets (Advanced Instruments, LLC, Norwood, MA, USA) during the three-month period of routine milking control. The milk component analysis was performed using the CombiFoss 7 DC automatic analyzer (FOSS Analytical A/S, Hilleroed, Denmark), which includes MilkoScan (near-infrared spectroscopy) and Fossomatic 7 DC (flow cytometry). The following parameters were determined: total solids (TS), milk solids not fat (MSNF), true protein (PT) and crude protein (PC) content, β-casein (Cas. β), fat content (Fat), fatty acids including saturated fatty acids (SFA), monounsaturated fatty acids (MUFA), polyunsaturated fatty acids (PUFA), long-chain fatty acids (LCFA), medium-chain fatty acids (MCFA), short-chain fatty acids (SCFA), myristic acid (C14:0), palmitic acid (C16:0), oleic acid (C18:1), trans-isomers of fatty acids (TFA), lactose, acetone, beta-hydroxybutyrate (BHB), urea, freezing point depression (FPD), and acidity (pH).

### 2.2. Genotyping and Quality Control of Data

The DNA was extracted using the DNA-Extran kit and following the manufacturer’s protocol (Syntol, Moscow, Russia). The qualitative and quantitative parameters of the extracted DNA were determined using a Qubit fluorometer (Invitrogen, Life Technologies, Waltham, MA, USA, www.invitrogen.com/qubit, accessed on 25 October 2023) and a NanoDrop 1000 instrument (Thermo Fisher Scientific Inc., Stoughton, MA, USA) according to the manufacturers’ instructions. For the study, samples with a ratio of A260/280 of 1.7–2.0 were used. Genotyping was performed using the Illumina GoatSNP53K BeadChip panel (Illumina Inc., San Diego, CA, USA) and the Illumina BeadStudio [V2.0]. The genotyping results were filtered using the standard methodology with the software RStudio 2023.03.0 (https://posit.co/download/rstudio-desktop/, accessed on 5 September 2023) and plink 1.9 (http://zzz.bwh.harvard.edu/plink/, accessed on 5 September 2023; Purcell et al., 2007 [22]).

The quality control and data filtering for each SNP and sample were performed using the plink 1.9 software package (http://zzz.bwh.harvard.edu/plink/, accessed on 5 September 2023), applying the following filters: individual sample call rate no less than 90% for all investigated SNPs (--mind); SNP call rate no less than 90% for all genotyped samples (--geno); minor allele frequency (MAF) greater than 0.01 or 0.05 (--maf 0.01); deviation of SNP genotypes from Hardy–Weinberg equilibrium in the tested samples with a significance level of *p*-value < 10^−6^ (--hwe). Additionally, an assessment of the linkage disequilibrium (LD estimation) was performed for the investigated SNPs with an r^2^ < 0.2 and a step size of 50 kb (--indep-pairwise). After data filtering, a total of 47155 SNPs were used in the analysis.

### 2.3. Genome-Wide Association Studies and Gene Analysis

To identify associations between the SNP markers and milk components, a multiple linear regression analysis implemented in plink 1.9 was used. To confirm the significant influence of the SNPs and determine significant regions in the genome of the studied goats, a Bonferroni test was performed with a *p*-value threshold of <1.06 × 10^−6^; 0.05/47,155 for genome-wide signification and a *p*-value threshold for suggestive associations of 2.12 × 10^−5^; 1/47,155. Manhattan plots illustrating the distribution of significant DNA polymorphisms across autosomes were generated using the ggplot2 package in RStudio [23]. 

To remove environmental and permanent effects and analyze the normal distribution of the studied traits, generalized linear models were applied using the STATISTICA 10 software.

The typical linear regression model in a genetic association study is:$$E\left[Y\right]=\beta_{0}+\beta_{X}X+\beta_{G}G,$$
where *β_G_* is the parameter of interest quantifying the association between a genotype G and the mean of an outcome *Y*. Further, *X* is a small set of *p* covariates, such as age and gender. Denote X=(1,X,G) and $\beta =\left(\beta_{0},\beta_{X},\beta_{G}\right)$. As long as the model for the mean of the outcome (1) is correct, the ordinary least squares estimator $\beta ={\left(X^{T}X\right)}^{-1}X^{T}$ is an unbiased estimator of *β*. It is a weighted average of the model outcome *Y* with weights that dependent on the covariate set *X*.

Gene identification based on SNP positions, as defined according to the ARS1.2 genome assembly, was performed using the Ensembl Genes web resource, release 103 [24]. Structural annotation of the genomic regions covering a window of ±0.20 Mb from the identified SNPs was conducted using the DAVID v6.8 program (https://david.ncifcrf.gov, accessed on 10 September 2023) [25].

## 3. Results

Descriptive statistics for the milk parameters of Karachai goats analyzed using infrared spectroscopy are presented in Table 1.

The analysis of the obtained data shows that the freezing point depression and acidity were characterized as the parameters with the least variability, with a coefficient of variation (Cv) not exceeding 4.2%. The coefficient of variation in the mass fraction of truth protein (PT) and crude protein (PC), as well as β-casein, was in the range of 23–26%, which does not contradict the data from other studies. They were followed by MSNF, lactose, TS, urea, polyunsaturated, saturated, and short-chain fatty acids with a Cv not exceeding 33.0%. The Cv of the other fatty acids ranged within 45.5%. Metabolites such as acetone and BHB demonstrated a great variability, with a Cv exceeding 330.0%.

The lack of variability in acidity may be due to the biological characteristics of goats of this breed and this aspect requires further research.

The range of variability in the content of dry matter, fat, protein, lactose, urea, and saturated fatty acids indicated a normal distribution of these traits in the studied sample. However, monounsaturated fatty acids showed greater variability, which may be related to the individual characteristics of goats in producing certain fatty acids, such as palmitic acid and oleic acid.

The high variability in the levels of acetone and BHB is likely due to the manifestation of hidden ketosis in some goats, leading to a significant increase in the concentration of these substances. For example, the acetone level in certain animals reached 1.62 mmol/L compared to the sample mean of 0.035 mmol/L, while BHB reached 2.36 mmol/L compared to the mean of 0.038 mmol/L. There is no available literature on the normal range of acetone and BHB content in goat milk. However, it is known that in cows, the normal levels are up to 0.72 mmol/L for acetone and 1.2 mmol/L for BHB, with an excess indicating subclinical ketosis [26]. 

Figure 1 presents the visualization results of the location of statistically significant polymorphic sites across 29 chromosomes for some of the milk parameters in Karachai goats. 

Associations were found for the TS parameter with 23 SNPs (5 highly significant and 11 suggestive), distributed across 11 chromosomes. For PT, PC, and Cas. β, 16 SNPs were associated, distributed across 10 chromosomes, with a genome-wide significance observed for 3, 6, and 2 SNPs, respectively, located on chromosomes 1, 2, 8, and 24.

For fat and SFA, associations were also found with 16 SNPs, localized on 12 chromosomes, with 2 and 3 SNPs, respectively, showing genome-wide significance on chromosomes 2, 3, 8, and 25. MUFA and PUFA were associated with 13 and 12 SNPs of suggestive significance, respectively, distributed across nine chromosomes. No genome-wide significant SNPs were identified for these parameters. PUFA, MCFA, and SCFA were associated with 12, 10, and 8 SNPs, respectively, distributed across 16 chromosomes. Among them, three and four SNPs showed genome-wide significance on chromosomes 2, 3, 8, 10, 16, and 25. 

Associations were found for lactose and urea content in Karachai goat milk with 12 and 8 SNPs, respectively, distributed across eight chromosomes (for lactose—1, 2, 6, 8, 11, 17, 18, and 21; for urea—2, 3, 8, 10, 15, 18, 19, and 27). Among them, six SNPs showed genome-wide significance for lactose content (on 1, 6, 8 chromosomes) and three SNPs for urea content (on 23 chromosome). Acetone content was associated with 21 SNPs, including 4 genome-wide significant SNPs distributed across chromosomes 3, 7, 9, and 27. Freezing point and acidity were associated with 18 and 16 SNPs, respectively, with 1 and 2 genome-wide significant SNPs on chromosomes 1 and 8 (Table 2).

Thus, the largest number of highly significant SNPs was identified on chromosomes 2 and 8, with some of them being associated with multiple traits. For instance, snp997-scaffold1026-378556 on chromosome 8 is associated with three traits (PC, PT, and lactose content), while snp43681-scaffold585-2255525 and snp33285-scaffold391-913110 are associated with two traits each (PT and lactose, PC and saturated fatty acids). Three SNPs—snp18646-scaffold1882-539299, snp8325-scaffold130-2860751, and snp8326-scaffold130-2909971 on chromosome 2—are associated with two traits each (dry matter content and saturated fatty acids; PC and Cas. β; PC and PT). Two polymorphisms—snp18646-scaffold1882-539299 and snp33285-scaffold391-913110, on chromosomes 2 and 8, respectively—correspond to five traits (saturated, medium-chain, and short-chain fatty acids and myristic and palmitic acid). Additionally, snp16908-scaffold1766-616140 on chromosome 25 should be noted, as it is shared among seven closely related traits: fat content; long-chain, medium-chain, and short-chain fatty acids; palmitic and oleic acid; and trans-isomers of fatty acids (Appendix A).

Of the 167 highly significant and subthreshold SNPs associated with the studied indicators of the composition and properties of Karachai goat milk, 10 SNPs were localized within the protein-coding genes and 33 SNPs in close proximity to them (±0.2 Mb). Functional annotation of all 43 SNPs in the biological library in the DAVID program (https://david.ncifcrf.gov, accessed on 5 September 2023) identified 66 significant candidate genes on 15 chromosomes (Figure 2) (Appendix A).

A total of 14 candidate genes associated with the protein and β-casein content in the milk of Karachai goats were identified, localized on six different chromosomes (Table 3). These genes were related to heart development and function (ADRA1A, NKX3-1, NKX2-6, STC1 on chromosome 8, ODAD2 on chromosome 13, CASP7 on chromosome 26), digestive tract development (NKX2-6), bone development (TNN on chromosome 16), muscle tissue development (NRAP on chromosome 26), protein metabolism processes (BAG2 on chromosome 23), lipid transport, and the maintenance of mitochondrial calcium ion homeostasis (PDZD8 on chromosome 26) (Table 3).

A total of 25 candidate genes associated with the fat content and fatty acid composition in the milk of Karachai goats were identified, distributed across 10 different chromosomes (Table 4).

The described functions of these candidate genes are related to lipid metabolism processes and adipose tissue development. For example, the INSIG1 gene (chromosome 4) is involved in triglyceride metabolism, cholesterol biosynthesis, metabolism and homeostasis, the regulation of fatty acid biosynthesis, and adipocyte differentiation. The PAXIP1 gene (chromosome 4) participates in adipose tissue development, the BAAT gene (chromosome 8) is involved in fatty acid metabolism, the PLPPR1 gene (chromosome 8) is involved in phospholipid metabolism, the LACTB gene (chromosome 10) regulates lipid metabolism processes, the FMO1 and FMO2 genes (chromosome 16) are involved in organic acid metabolism, the ECI1 gene (chromosome 25) is involved in the beta-oxidation of fatty acids, the PGP gene is involved in glycerol biosynthesis, and the ABCA3 gene (chromosome 25) regulates cholesterol efflux, phosphatidylcholine and phosphatidylglycerol metabolic processes, lipid biosynthesis regulation, and phospholipid homeostasis.

Among all the identified genes, ECI and PGP should be highlighted, as they are associated with the beta-oxidation of fatty acids and glycerol biosynthesis, making them candidate genes for the fatty acid composition in the milk of Karachai goats.

These genes should be considered potential DNA markers for the fatty acid composition of Karachai goat milk.

A total of 17 candidate genes associated with the acetone, BHB, and urea content in the milk of Karachai goats were identified, distributed across six chromosomes (Table 5). The described functions of these candidate genes are related to neurogenesis (ETV6 on chromosome 2 and NTN5 on chromosome 18), immune response (ATF2 on chromosome 2, INPP5D on chromosome 3, CDK6 on chromosome 4, CACNA1C on chromosome 5), the development of the heart, blood vessels, and brain (ATF2 on chromosome 2, CACNA1C on chromosome 5, SPHK2 on chromosome 18), cholesterol metabolism (SULT2B1 on chromosome 18) and carbohydrate and glycoprotein metabolism (NEU2 on chromosome 3, FUT1 on chromosome 18) (Appendix A).

The summary of data on the significant biological functions of candidate genes allows us to conclude that the largest number—29 genes, or 37.7%—was associated with the development of physiological traits, including response to the external environment (GO:0003298). 

Next 17 genes, or 22.0%, regulate metabolic process (GO:0008152) and cellular processes (GO:0005575) and also the development of the nervous–muscular system (GO:0050877, GO:0003012). Eight genes, or 10.4%, were associated with the regulation of tissue–skeletal (GO:0009888, GO:0035989) and vascular framework (GO:0001944) formation. 

The smallest number of genes (four genes or 5.2%) was associated with intrauterine (embryonic) development (GO:0009790) and the same number of genes was associated with immunity (GO:0006955) (Appendix A).

## 4. Discussion

Goats, as subjects of genetic research, are increasingly attracting the attention of scientists. With the development of the GoatSNP50 BeadChip by Illumina Inc. [27], GWASs (genome-wide association studies) became available for goats of different productivity directions [9,21]. Through these studies, it has been possible to identify certain genes associated with various economically important traits in goats, such as fiber productivity, including fiber color [28,29,30], meat productivity traits [31,32,33], reproduction [34,35,36], and adaptation ability [37,38,39]. Several studies have focused on identifying genes associated with milk production traits in goats, as well as exterior traits that have a greater impact on level of their development [40,41,42]. To further expand research in this direction for dairy goats, work is being undertaken to develop a new panel of liquid SNP chips at a lower cost, making them more accessible for both scientific research and future practical breeding. This panel contains 54,188 SNPs based on genotyping technology using targeted sequencing (GBTS) [43].

Most of the research in dairy goat breeding has been carried out on goats with pronounced milk productivity: Saanen, Alpine, and Toggenburg breeds. In Russia, dairy goat farming, as well as throughout the world, is largely based on the use of goats of these transboundary breeds; however, in certain regions, for example, the North Caucasus, locally selected goats, which include Karachai goats, are becoming more common. It should be noted that for dairy–meat goat breeds in general, and local breeds in particular, not enough genetic research has been carried out, which justifies the relevance of this work.

The objective of the research was to search for candidate genes associated with traits related to milk production in Karachai goats. To determine candidate genes, all identified SNPs were used, both genome-wide (highly significant) and subthreshold significance (suggestive). The largest number of highly significant SNPs were identified on chromosomes 2 and 8, with some SNPs associated with multiple traits. Thus, two SNPs on chromosomes 2 and 8 were shared by protein, β-casein, lactose, and saturated fatty acids. One SNP each on chromosomes 8 and 5 is common to closely related traits such as fat content, all classes of fatty acids, and their trans-isomers.

Comparing our data with the results of other studies, it should be noted that the established SNPs are located on different chromosomes; however, data from other authors have established a connection between SNPs and not one trait but multiple traits. Thus, Scholtens M. et al., (2020) [17] identified 43 SNPs, of which the largest number—31—was located on 19 chromosomes, while 7 were associated with several traits of milk productivity of goats, 16 with milk yield, fat increase, and protein in milk, and 8 with one or two traits. Martin et al., (2018) [14] and Taluarn, E. et al., (2020) [15] also found more “selection signatures” on chromosome 19 than on other dairy goat chromosomes, with some SNPs associated with multiple traits. Other studies found that most of the SNPs associated with increased fat in goat milk other than chromosome 19 were located on chromosomes 14 and 29. Massender E et al. [18] identified 189 unique SNPs occurring on all chromosomes, but regions on chromosomes I6 (86,050,088 and 86,858,026 bp) and CHI14 (80,143,561, 81,347,395 and 81 658,383 bp) were the most significant to the milk composition traits.

Genomic identification of the currently found SNPs associated with the addition of protein and casein and functional annotation of the candidate genes revealed their connection with heart function (*ADRA1A*, *NKX3-1*, *NKX2-6*, *STC1*, *ODAD2*, *CASP7*), the alimentary canal (NKX2-6), the development of bone (*TNN*) and muscle tissue (*NRAP*), and protein metabolism processes (*BAG2*). 

In studies by Signer-Hasler H. et al. [44], the STC1 gene, determined using the ROH method, was identified as a novel domestication gene affecting important traits such as body size and milk production in Swiss goats. According to its biological function, it is associated with the contraction of the cardiac muscle cells (group 7 of this study) and milk production, and it can be assumed to be a special marker gene for dairy goats adapted to mountainous areas.

The other genes we identified were not described in studies by other authors as potential genetic markers of the milk protein composition. Results obtained by other researchers on Murciano-Granadina, Norwegian and Sarda goats demonstrated the association of other genes (CSN1S1, CSN1S2, CSN2, CSN3, as well as α-LA) with milk yield and milk composition (amount of protein, fat, dry matter, lactose and somatic cells) [45,46,47].

In several other studies, the polygenic nature of the influence of individual genes on the compositional characteristics of goat milk is noted. For example, the *AGPAT6* gene (1-acylglycerol-3-phosphate-O-acyltransferase) is considered to be associated with both decreased milk production and increased milk protein and fat content [48]. The *PRL* (prolactin) and *PRLR* (prolactin receptor) genes are associated with lactose, protein, and fat content [49]. The *ACACA* gene (acetyl-CoA carboxylase alpha) is involved in the regulation of enzymes involved in fatty acid biosynthesis and is associated with milk protein and fat content [50]. The *SCD* gene (Stearoyl-CoA desaturase), which is involved in the synthesis of monounsaturated fatty acids in the adipose tissue and mammary glands, directly influences the profile of other fatty acids in goat milk [51]. The *PPARγ* gene (Peroxisome Proliferator-activated Receptor Gamma) encodes a protein that has a high impact on the transcription of genes such as *LPL*, *FASN*, *ACACA*, *SCD*, *PLIN2*, *PLIN3*, *FABP3*, and *PNPLA2*, which are involved in lipid metabolism [52]. In turn, the *SCD* gene alters the composition of long-chain fatty acids (LCFA), and its expression is directly regulated by *SREBP-1* and *PPARγ1* in lactating goat mammary gland cells [53].

A comparison of our data with the results of other researchers made it possible to identify analogies in relation to such genes as INSIG1, BAAT, and LACTB. A study of the INSIG1 gene and its expression showed that it is one of the genes that controls the synthesis of milk fat in goats’ mammary gland cells [54]. In our studies, the INSIG1 gene was identified in structural indicators such as polyunsaturated and long-chain fatty acids and oleic acid, and, within it, there is a reliably obtained SNP (snp1935-scaffold1053-1528396).

Analysis of the BAAT gene in a population of Liaoning cashmere goats revealed that the GG genotype is associated with cashmere characteristics and the AG genotype is associated with body size and milk production traits [55]. In our studies, concerning Karachai goats, the BAAT gene is associated with the mass fraction of fat in the milk.

In our study, we also identified 12 genes that exhibited a polygenic influence. Three-quarters of these genes were simultaneously associated with the dry matter content and fatty acids. These genes include *METTL*, *SLC1A8*, *PHACTR1*, *FMO2*, *ECI1*, *PGP*, *ABCA3*, and *AMDHD2*, which influenced the content of dry matter and polyunsaturated fatty acids (PUFA), saturated fatty acids (SFA), monounsaturated fatty acids (MUFA), trans-isomers of fatty acids, C14, and C16. Other genes were associated with different traits. For example, the AGTPBP1 gene was associated with PC, β-casein, lactose, and pH; the *TNN* gene was associated with PT and PC; the *ATF2* gene was associated with BHB and acetone; and the *NECAP2* gene was associated with TS and urea.

## 5. Conclusions

Summarizing the above, it should be noted that the genes identified in our study related to milk components in Karachai goats, which are dual-purpose goats, differed from those identified in relation to milk in dairy goats. Apparently, the genetic architecture in goats with different levels of the main productivity trait, milk production, is different. Additional research on larger populations and different breeds is necessary for a better understanding of the genetic mechanisms underlying milk production in goats.

## Figures and Tables

**Figure 1 animals-14-00327-f001:**
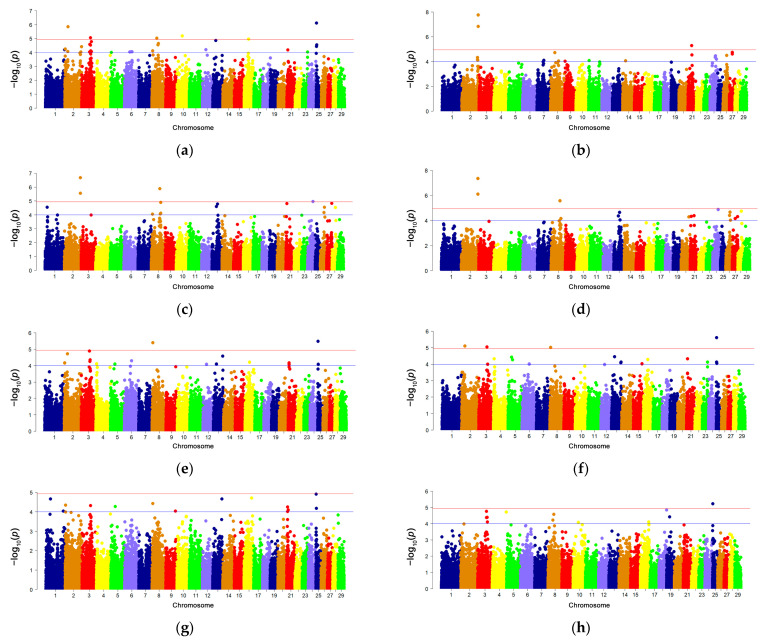
The distribution of statistically significant SNPs in 29 autosomes of Karachai goats for the following traits: (**a**)—TS; (**b**)—MSNF; (**c**)—PT; (**d**)—Cas. β; (**e**)—Fat; (**f**)—SFA; (**g**)—MUFA, (**h**)—PUFA; (**i**)—LCFA; (**j**)—MCFA; (**k**)—SCFA; (**l**)—TFA; (**m**)—C14:0; (**n**)—C16:0, (**o**)—C18:1; (**p**)—Lactose; (**q**)—Acetone; (**r**)—BHB; (**s**)—Urea; (**t**)—FPD. The logarithm of the q value (Y-axis) is plotted for each chromosome (X-axis). On the Y-axis, the lower line represents a significance level of *p* ≤ 0.00001, and the upper line represents a significance level of *p* ≤ 0.000001.

**Figure 2 animals-14-00327-f002:**
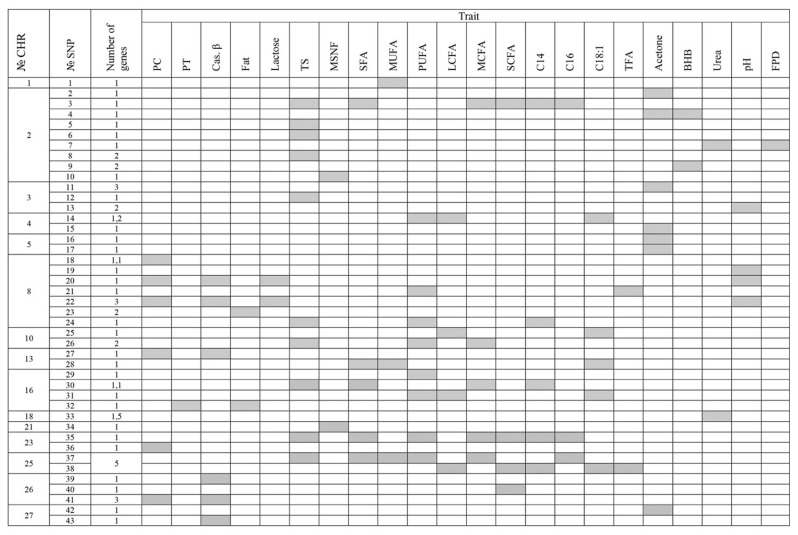
Single nucleotide polymorphisms (SNPs) located within or in close proximity to protein-coding genes, associated with compositional and milk property indicators in Karachai goats. Note: No. CHR—goat chromosome number, No. SNP—sequential number of SNPs; traits: PC—crude protein; PT—true protein; Cas. β—β-casein; Fat—fat content; TS—total solids; MSNF—milk solids not fat; SFA—fatty acids including saturated fatty acids; MUFA—monounsaturated fatty acids; PUFA—polyunsaturated fatty acids; LCFA—long-chain fatty acids; MUFA—medium-chain fatty acids; SCFA—short-chain fatty acids; C14:0—myristic FA; C16:0—palmitic FA; C18:1—oleic FA; TFA—trans-isomers of fatty acids; BHB—beta-hydroxybutyrate; FPD—freezing point depression; pH—acidity.

**Table 1 animals-14-00327-t001:** Descriptive statistics for milk parameters in the studied sample of Karachai goats ^1^.

Trait	Abbreviated Name	Min	Max	Mean	Std. Dev	C_v_, %
TS and MSNF, %	TS	7.44	24.59	14.58	0.18	19.31
	MSNF	5.32	15.63	9.44	0.09	12.63
Protein and β-casein, %	PT	3.07	10.53	4.45	0.07	25.47
	PC	3.22	11.03	4.67	0.07	26.22
	Cas. β	2.10	9.27	3.62	0.06	23.85
Fat, % FA, TFA, g/100 g	Fat	2.37	10.13	5.68	0.12	31.70
	SFA	1.36	8.85	3.92	0.19	32.73
	MUFA	0.487	3.412	0.337	0.077	45.2
	PUFA	0.042	0.520	0.228	0.009	31.5
	LCFA	0.073	4.675	1.889	0.114	47.2
	MCFA	0.011	4.689	2.223	0.111	38.9
	SCFA	0.008	2.390	1.127	0.061	42.1
	C14:0	0.150	1.368	0.593	0.032	41.8
	C16:0	0.305	3.138	1.348	0.076	43.8
	C18:1	0.016	3.604	1.462	0.085	45.5
	TFA	0.001	0.493	0.185	0.013	53.8
Lactose, %	Lactose	0.430	4.940	4.02	0.05	17.15
Metabolites, mmol/L, mg × 100 mL^−1^ *	Acetone	−0.51	1.62	0.035	0.01	334.13
	BHB	−0.40	2.36	0.038	0.01	397.22
	Urea	15.70	103.7	71.06	0.95	19.38
Technological properties of milk, −1 × 10^−3^ °C **	FPD	483.0	612.0	546.25	1.40	2.24
	pH	5.02	6.82	6.39	0.02	4.18

^1^ *Note*. Traits: TS—total solids, MSNF—milk solids not fat, PT—true protein, PC—crude protein, Cas. Β—β-casein, Fat—fat content, SFA—saturated fatty acids, MUFA—monounsaturated fatty acids, PUFA—polyunsaturated fatty acids, LCFA—long-chain fatty acids, MCFA—medium-chain fatty acids, SCFA—short-chain fatty acids, C14:0—myristic acid, C16:0—palmitic acid, C18:1—oleic acid, TFA—trans-isomers of fatty acids, BHB—beta-hydroxybutyrate, FPD—freezing point depression, pH—acidity. * units for urea; ** units for FPD.

**Table 2 animals-14-00327-t002:** Number of SNPs significantly associated with milk composition parameters of Karachai goats ^1^.

Trait	Genome-Wide Threshold		Suggestive Threshold	
	Number	Chr	Number	Chr
TS	5	2, 3, 8, 10, 25	18	1, 2, 3, 4, 6, 8, 12, 13, 16, 23, 25
MSNF	2	2, 21	15	2, 7, 8, 9, 11, 14, 21, 24, 27
PT	3	2, 8	13	1, 2, 8, 13, 21, 24, 26, 27, 28
PC	6	1, 2, 8, 24	13	1, 2, 8, 13, 21, 23, 26, 27, 28
Cas. β	2	2	16	8, 13, 20, 21, 24, 26, 27, 28
Fat	2	8, 25	14	2, 3, 4, 5, 6, 12, 13, 16, 21, 25
SFA	3	2, 3, 8	13	4, 5, 6, 13, 15, 16, 21, 23, 25
MUFA	-		13	1, 2, 3, 8, 9, 13, 16, 21, 25
PUFA	-		12	3, 4, 8, 10, 16, 18, 19, 23, 25
LCFA	3	8, 16, 25	9	1, 2, 3, 4, 8, 9, 10, 21, 29
MCFA	4	2, 3, 10, 25	6	13, 15, 16, 18, 21, 23
SCFA	4	2, 3, 8, 25	4	3, 6, 23, 25
C14:0	3	2, 3, 8	6	5, 16, 18, 23, 25, 27
C16:0	3	2, 3, 25	9	5, 6, 15, 18, 21, 23
C18:1	3	8, 16, 25	11	2, 3, 4, 9, 10, 13, 21, 25
TFA	1	25	4	3, 8, 18
Lactose	6	1, 6, 8,	6	2, 6, 11, 17, 18, 21
Acetone	4	3, 7, 9, 27	17	2, 4, 5, 6, 7, 10, 17, 27
BHB	1	2	6	2, 3, 4, 6
Urea	3	2, 3	5	8, 10, 15, 18, 19, 27
FPD	2	1	16	1, 2, 3, 4, 11, 13, 15, 24
pH	4	1, 8	12	1, 3, 13, 17, 21

^1^ *Note*. Traits: TS—total solids, MSNF—milk solids not fat, PT—true protein, PC—crude protein, Cas. Β—β-casein, Fat—fat content, SFA—saturated fatty acids, MUFA—monounsaturated fatty acids, PUFA—polyunsaturated fatty acids, LCFA—long-chain fatty acids, MCFA—medium-chain fatty acids, SCFA—short-chain fatty acids, C14:0—myristic acid, C16:0—palmitic acid, C18:1—oleic acid, TFA—trans-isomers of fatty acids, BHB—beta-hydroxybutyrate, FPD—freezing point depression, pH—acidity.

**Table 3 animals-14-00327-t003:** Candidate genes associated with protein and β-casein content in the milk of Karachai goats ^1^.

Traits	№ Chr	№ SNP	SNP	Gene/Position
PC	8	18	snp10589-scaffold1376-2594525^73806016…73406016^	* **DPYSL2** ^73547764…73663711^ *
				*ADRA1A^73747397…73862840^*
PC, Cas. β	8	20	snp43681-scaffold585-2255525^78696104…78296104^	*AGTPBP1^78600749…78791306^*
PC, Cas. β	8	22	snp997-scaffold1026-378556^70884382…70484382^	*NKX3-1^70600953…70606837^*
				*NKX2-6^70639383…70643794^*
				*STC1^70791218…70806076^*
PC, Cas. β	13	27	snp5221-scaffold1180-236240^36057527…35657527^	*ODAD2^35971967…36143096^*
PT	16	32	snp8683-scaffold131-4589642^54997494…54597494^	*TNN^54831011…54903426^*
PC	23	36	snp48737-scaffold692-158314^45730320…45330320^	*BAG2^45658115…45668381^*
Cas. β	26	39	snp18573-scaffold1878-337881^14293322…13893322^	*PDZD8^14290884…14368219^*
PC, Cas. β	26	41	snp47577-scaffold67-3351554^17989628…17589628^	*CASP7^17808414…17850585^*
				*NRAP^32471325…32547298^*
				*HABP2^17935249…17970574^*
Cas. β	27	43	snp55772-scaffold864-4012988^20994598…20594598^	*MFHAS1^20648690…20757838^*

^1^ *Note*. Traits: PC—crude protein; PT—true protein; Cas. β—β-casein; № Chr—goat chromosome number; № SNP—sequential number of SNP in Figure 1; SNP—name of the reliable SNP and its position in the genome (indicated as superscript numbers); gene—gene within or in close proximity to which the reliable SNP is localized (genes with localized SNPs are shown in bold).

**Table 4 animals-14-00327-t004:** Candidate genes associated with fat content and fatty acid composition in the milk of Karachai goats ^1^.

Traits	№ Chr	№ SNP	SNP	Gene/Position
MUFA	1	1	snp27412-scaffold292-177570^141251692…140851692^	*MX2* ^141193475…141232778^
SFA, MCFA, SCFA, C14, C16	2	3	snp18646-scaffold1882-539299^111252769…110852769^	*METTL8^111185392…111270066^*
PUFA, LCFA, C18:1	4	14	snp1935-scaffold1053-1528396^2946300…2546300^	** *INSIG1^2747017^* ** * ^…2757803^ *
				*EN2^2659430…2666542^*
				*PAXIP1^2922811…2963751^*
PUFA, TFA	8	21	snp47195-scaffold66-1841919^47980184…47580184^	*CEMIP2^47675207…47756816^*
Fat	8	23	snp34748-scaffold412-871719^91227795…90827795^	*BAAT^90845022…90861305^*
				*PLPPR1^90695441…90835782^*
PUFA, C14	8	24	snp28090-scaffold300-3913146^39951214…39551214^	*SLC1A^39770036…39845247^*
LCFA, C18:1	10	25	snp1441-scaffold104-780285^25686826…25286826^	** *FUT8* ** * ^25310706…25629694^ *
PUFA, MCFA	10	26	snp24669-scaffold251-1625913^56289166…55889166^	*TPM1*
				*LACTB*
SFA, MUFA, C18:1	13	28	snp8522-scaffold1308-1729010^75567491…75167491^	*EYA2^75025156…75283017^*
PUFA	16	29	snp18361-scaffold186-251734^68574935…68174935^	*PROX1^68179003…68234777^*
SFA, MCFA, C14	16	30	snp50562-scaffold727-524836^37486538… 37086538^	** *FMO2^37286455^* ** * ^…^ * * ^37330116^ *
				*FMO1^37336303…37376884^*
MUFA, LCFA, C18:1	16	31	snp3754-scaffold112-3973504^63571269…63171269^	*NCF2^63224955…63263716^*
Fat	16	32	snp8683-scaffold131-4589642^54997494…54597494^	*TNN^54831011…54903426^*
SFA, MCFA, PUFA, SCFA, C14, C16	23	35	snp10273-scaffold1368-2701834^7104785…6704785^	** *PHACTR1^6556030^* ** * ^…7073500^ *
SFA, MUFA, PUFA, MCFA, LCFA, SCFA, C16, C18:1, C14, TFA	25	37, 38	snp16907-scaffold1766-582489^2027754…1627754^ snp16908-scaffold1766-616140^2061790…1661790^	*ECI1^1672290…1685556^*
				*PGP^1649864…1652731^*
				*ABCA3^1701870…1743075^*
				*AMDHD2^1915270…1921956^*
				*PDPK1^1932902…1996678^*
SCFA	26	40	snp41130-scaffold532-1727870^30209081…29809081^	*SEMA4G^29853719…29867663^*

^1^ *Note*. Traits: Fat—fat content; SFA—saturated fatty acids; MUFA—monounsaturated fatty acids; PUFA—polyunsaturated fatty acids; LCFA—long-chain fatty acids; MCFA—medium-chain fatty acids; SCFA—short-chain fatty acids; C14:0—myristic acid; C16:0—palmitic acid; C18:1—oleic acid; TFA—trans fatty acids; № Chr—goat chromosome number; № SNP—sequential number of SNP in Figure 1; SNP—name of the reliable SNP and its position in the genome (indicated as superscript numbers); gene—gene within or in close proximity to which the reliable SNP is localized (genes with localized SNPs are shown in bold)**.**

**Table 5 animals-14-00327-t005:** Candidate genes associated with acetone, BHB, and urea content in the milk of Karachai goats ^1^.

Traits	№ Chr	№ SNP	SNP	Gene/Position
BHB, Acetone	2	4	snp9402-scaffold1341-2132092^114722555…114322555^	** *ATF2* ** * ^114518050^ * * ^…^ * * ^114597771^ *
Urea	2	7	snp25937-scaffold2682-105434^814121…414121^	*NECAP2^486514^* * ^…^ * * ^500217^ *
BHB	2	9	snp2511-scaffold1071-129957^15102755…14702755^	*HPCA^14931533^* * ^…^ * * ^14940116^ *
				*FNDC5^14907902^* * ^…^ * * ^14915966^ *
Acetone	3	11	snp22179-scaffold219-771742^8377011…7977011^	*NEU2^8043799^* * ^…^ * * ^8057992^ *
				*GIGYF2^8226338^* * ^…^ * * ^8358229^ *
				*INPP5D^7877943^* * ^…^ * * ^8015888^ *
	4	15	snp38614-scaffold49-1209913^110877395… 110477395^	*CDK6^110413189^* * ^…^ * * ^110678573^ *
	5	16	snp47348-scaffold666-52899^97220952… 96820952^	*ETV6^96570433^* * ^…^ * * ^96857059^ *
		17	snp12158-scaffold1450-249468^107408739…107008739^	*CACNA1C^107328368^* * ^…^ * * ^107719644^ *
Urea	18	33	snp18289-scaffold1857-308667^56470790…56070790^	** *SULT2B1^56239547^* ** * ^…^ * * ^56275810^ *
				*GRIN2D^56117047^* * ^…^ * * ^56151568^ *
				*SPHK2^56292056^* * ^…^ * * ^56299987^ *
				*NTN5^56321271^* * ^…^ * * ^56326460^ *
				*IZUMO1^56393899^* * ^…^ * * ^56396881^ *
				*FUT1^56399123^* * ^…^ * * ^56402123^ *
Acetone	27	42	snp51881-scaffold762-2349319^4343759…3943759^	*THRB^3763978^* * ^…^ * * ^4201001^ *

**^1^***Note*. Traits: BHB—beta-hydroxybutyrate; № Chr—goat chromosome number; № SNP—sequential number of SNP in Figure 1; SNP—name of the reliable SNP and its position in the genome (indicated as superscript numbers); gene—gene within or in close proximity to which the reliable SNP is localized (genes with localized SNPs are shown in bold).

## Data Availability

Data are contained within the article.

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
