# Peer review of "Genome-Wide Association Study of Milk Composition in Karachai Goats"

_animals, 2024, doi:10.3390/ani14020327_

Round 1

Reviewer 1 Report

Comments and Suggestions for Authors

Line 63 define the acronym SNP before using it

Line 125 Assotiation should be Association

Line 134 Please specify what Generalized Linear Models you have used and the link functions associated.

Table 1. Please make sure tables are not broken between pages. Use footnotes to explain any acronym used even if explained before in the text. also check in other tables

Line 157 be specific,… which fatty acids

Figure 1. Could you please try to fit all the figure in 1 page. You could do it by showing 3 graphs per line. Make sure you increase the font size as well. Make sure the legend is kept in the same page also.

Table 3 broken. Some traits are also half missing

Table 4 broken

Table 5 broken

In all the result section please specify models used to find associations and state p-values for significances.

Invasive tissue and blood sample collection. Ethics approval?

Author Response

Dear Colleague,

We have carefully studied all the issues that you outlined in your letter and we would like to inform you of the following:

Line 63 define the acronym SNP before using it

We have made changes according to your comments.

Line 125 Assotiation should be Association

We have made changes according to your comments.

Line 134 Please specify what Generalized Linear Models you have used and the link functions associated.

The used Generalized Model was added to the material and methods.

Table 1. Please make sure tables are not broken between pages. Use footnotes to explain any acronym used even if explained before in the text. also check in other tables

Line 157 be specific,… which fatty acids

We have made changes according to your comments.

Figure 1. Could you please try to fit all the figure in 1 page. You could do it by showing 3 graphs per line. Make sure you increase the font size as well. Make sure the legend is kept in the same page also.

We tried to follow your suggestion.

Table 3 broken. Some traits are also half missing

We have made changes according to your comments.

Table 4 broken

We have made changes according to your comments.

Table 5 broken

We have made changes according to your comments.

In all the result section please specify models used to find associations and state p-values for significances.

 We have made changes according to your comments.

Invasive tissue and blood sample collection. Ethics approval?

A copy of our university's ethics committee approval is attached and we added the information about it below the Funding.

In the text of the manuscript, all changes are highlighted in green.

Once again please accept our gratitude for your support and assistance.

Reviewer 2 Report

Comments and Suggestions for Authors

Dear Dr Rita Meng   

Mananging Editor of Animals

The work entitled "Genome-wide association study of milk composition 2 in Karachai goats" is interesting but has some problems with understanding.

The abstract is quite indicative of what has been done in the study.

The introduction is not very precise and does not explain well the scientific reason why the test is being conducted.

M&Ms are eaustive and easy to interpret. The statistical analysis is approximate and it is not clear what has been done, we need to improve the formulation of what has been done.

The results contain an infinite amount of data that is difficult to manage and understand. You should try to focus exclusively on what is important.

The discussion is excessively long and dispersive without clarifying the results obtained.

Some assumptions are a little off base and not supported by data

The manuscript, although interesting, is too confusing and should be entirely rewritten to be easily readable, interpretable and repeatable.

Kind regards

Author Response

Dear Colleague,

First of all, please accept our sincere gratitude for your comments.

This is in no way an excuse, only explanation, but we tried to present the material based on the structure of already published manuscripts similar in topic and focus to our research (doi.org/10.21203/rs.3.rs-2966814/v1, DOI: 10.3168 /jds.2022-22223, doi.org/10.5194/aab-65-145-2022 and so on).

We agree, that the article probably contains too much information, but we needed to describe the indicators that were planned in the study as part of our grant. For our country, this is one of the first such studies and we tried to show as much as possible the results obtained.

We have made corrections to the text of the manuscript, taking into account your comments and the comments of the second reviewer, trying to make clarifications.

We really hope that the results of our work will be published; this is very important for our scientific team and the university.

Once again please accept our gratitude for your support and assistance.

Round 2

Reviewer 2 Report

Comments and Suggestions for Authors

Dear Mr Zeng

Animal Editorial Office 
The manuscript entitled "Genome-wide association study of milk composition in Karachai goats" has improved although it remains quite scattered due to the large amount of data presented. Certainly, to confirm the effect of these SNPs we would need to investigate better and with a larger number of animals.

The abstract does not improve the situation as it lists a very high number of chromosomes and genes that are associated with different physiological functions and therefore maintains the confusion.

The introduction is oriented towards only a part of the investigations carried out on goats and leaves out others, e.g. it does not talk about the knowledge obtained about proteins.

The M&Ms are comprehensive and the statistics are now more understandable. The results are presented quite well although too much data leads to several confusions. What is surprising is that variability is found only for the beta proteins and none for the other caseins which are the most variable as reported in the bibliography on the subject.

The discussion is incomplete and at times confusing.

The conclusions are in agreement with the aim and results but do not agree with what is written in the abstract.

The study is interesting mainly because it provides knowledge on a breed of goat that exists in a particular territory. In fact, these data can give inspiration to intensify research to improve this breed that lives in these difficult or marginal areas.

Kind regards

Author Response

Dear Colleague,

First of all, please accept our sincere gratitude for your comments.

We have really tried to correct the text of the manuscript in accordance with your comments, as far as possible.

The manuscript entitled "Genome-wide association study of milk composition in Karachai goats" has improved although it remains quite scattered due to the large amount of data presented. Certainly, to confirm the effect of these SNPs we would need to investigate better and with a larger number of animals.

The abstract does not improve the situation as it lists a very high number of chromosomes and genes that are associated with different physiological functions and therefore maintains the confusion.

We have rewritten the text to clarify the content of the abstract.

The introduction is oriented towards only a part of the investigations carried out on goats and leaves out others, e.g. it does not talk about the knowledge obtained about proteins.

The M&Ms are comprehensive and the statistics are now more understandable. The results are presented quite well although too much data leads to several confusions. What is surprising is that variability is found only for the beta proteins and none for the other caseins which are the most variable as reported in the bibliography on the subject.

The description about proteins variability was changed.

The discussion is incomplete and at times confusing.

We removed unnecessary descriptions that related to functional annotations, as well as those places in the text that duplicated information.

The conclusions are in agreement with the aim and results but do not agree with what is written in the abstract.

We tried to coordinate the texts of the conclusion and abstract.

Kind regards,

Once again please accept our gratitude for your support and assistance.
